# Methionine Adenosyltransferase I/III Deficiency Detected by Newborn Screening

**DOI:** 10.3390/genes13071163

**Published:** 2022-06-27

**Authors:** Vanessa Hübner, Luciana Hannibal, Nils Janzen, Sarah Catharina Grünert, Peter Freisinger

**Affiliations:** 1Department of Pediatrics, Metabolic Disease Center, Klinikum Reutlingen, Steinenbergstr. 31, 72764 Reutlingen, Germany; freisinger_p@klin-rt.de; 2Department of General Pediatrics, Adolescent Medicine and Neonatology, University Medical Center, Faculty of Medicine, University of Freiburg, Mathildenstraße 1, 79106 Freiburg, Germany; luciana.hannibal@uniklinik-freiburg.de; 3Screening Laboratory Hannover, Box 91 10 09, 30430 Hannover, Germany; n.janzen@metabscreen.de; 4Division of Laboratory Medicine, Centre for Children and Adolescents, Kinder- und Jugendkrankenhaus Auf der Bult, Janusz-Korczak-Allee 12, 30173 Hannover, Germany; 5Department of Clinical Chemistry, Hannover Medical School, Carl-Neuberg-Str. 1, 30625 Hannover, Germany

**Keywords:** MAT1A, hypermethioninemia, methionine adenosyltransferase, newborn screening, CNS symptoms

## Abstract

Methionine adenosyltransferase I/III deficiency is an inborn error of metabolism due to mutations in the *MAT1A* gene. It is the most common cause of hypermethioninemia in newborn screening. Heterozygotes are often asymptomatic. In contrast, homozygous or compound heterozygous individuals can develop severe neurological symptoms. Less than 70 cases with biallelic variants have been reported worldwide. A methionine-restricted diet is recommended if methionine levels are above 500–600 µmol/L. In this study, we report on a female patient identified with elevated methionine concentrations in a pilot newborn screening program. The patient carries a previously described variant c.1132G>A (p.Gly378Ser) in homozygosity. It is located at the C-terminus of MAT1A. In silico analysis suggests impaired protein stability by β-turn disruption. On a methionine-restricted diet, her serum methionine concentration ranged between 49–605 µmol/L (median 358 µmol/L). Her clinical course was characterized by early-onset muscular hypotonia, mild developmental delay, delayed myelination and mild periventricular diffusion interference in MRI. At 21 months, the girl showed age-appropriate neurological development, but progressive diffusion disturbances in MRI. Little is known about the long-term outcome of this disorder and the necessity of treatment. Our case demonstrates that neurological symptoms can be transient and even patients with initial neurologic manifestations can show normal development under dietary management.

## 1. Introduction

Methionine (Met) is a well-established newborn screening (NBS) marker for the identification of various inborn errors of metabolism, such as cystathionine-β-synthase deficiency (OMIM # 236200; CBS deficiency), glycine N-methyltransferase deficiency (OMIM # 606664; GNMT deficiency) or methionine adenosyltransferase I/III deficiency (OMIM # 250850; Mudd’s disease, MAT I/III deficiency) [1]. However, Met concentration may also be elevated secondary to liver disease or prematurity [1,2]. The human genome possesses three isoforms of methionine adenosyltransferase (MAT; EC 2.5.1.6): MAT1A is highly expressed in the liver, while MAT2A and MAT2B are expressed ubiquitously [3]. The most common cause of hypermethionemia in NBS is MAT I/III deficiency caused by mono- or biallelic variants in the *MAT1A* gene that encodes the two hepatic MAT isozymes I (homotetramer) and III (homodimer) [2,4,5]. MAT I/III catalyzes the formation of S-adenosyl methionine (AdoMet) from Met and adenosine triphosphate (ATP). AdoMet is required as a methyl donor for many trans-methylation reactions catalyzed by >200 methyltransferases in the body, for example in the biosynthesis of various neurotransmitters such as adrenaline or acetylcholine [6]. If MAT I/III enzyme activity is only partially reduced, the metabolic disorder is usually benign (ORPHA: 168598), whereas a complete loss of enzyme activity leads to severe neurological phenotypes with developmental delay, hyperreflexia and dystonia [2,4,6]. Myelination delay can be seen in the MRI [4]. Although the majority of affected individuals, in particular heterozygotes, are asymptomatic, patients with biallelic variants can show neurological symptoms [4,7]. Approximately 80–90% of cases identified by NBS are heterozygotes [8]. The incidence of MAT I/III deficiency is estimated to be 1:30,000 to 1:105,000, according to NBS data [7,8,9].

A low Met diet is usually recommended if Met levels are above 500–600 µmol/L, but the benefits of dietary treatment have not yet been fully established [4,10,11]. There is also no consensus on the therapeutic use of AdoMet [11]. Due to the reduced enzyme activity of MAT I/III, one would theoretically expect a reduced availability of AdoMet in biallelic patients. In the few cases examined, this could not be confirmed so far. The untreated patients showed both normal and increased AdoMet levels [4]. Independently of AdoMet levels, Chien et al. reported patients with CNS abnormalities on MRI that improved or even normalized with AdoMet supplementation. Supplementation could also be a treatment option for patients on a Met-restricted diet with decreased levels of AdoMet.

## 2. Materials and Methods

Crystallographic data of human MAT1A was downloaded from the Protein Data Bank under accession number 2OBV [12]. The complete structural analysis of wild-type human MAT1A can be found in its associated publication [12]. Images of human recombinant MAT1A, as well as in silico mutagenesis to recreate variant Gly378Ser and structural alignment of wild-type and mutated proteins, were created with PyMOL for Mac (PyMOL™ version 2.4.0, Schrödinger, LLC, New York, NY, USA).

## 3. Results

### 3.1. Case Presentation

The patient is the 4th child of consanguineous Pakistani parents. Two older siblings are healthy; one died in the newborn period from an unknown cause. The patient was born after an uneventful pregnancy in the 38th week of gestation. The patients’ parents agreed to participate in an NBS pilot study that included the measurement of methionine. The first NBS result showed an elevated level of Met (144 µmol/L, reference range 8–50 µmol/L). The control on day 16 was suggestive of homocystinuria with elevated total homocysteine (tHcy; 22.6 µmol/L, reference range < 13 µmol/L) and Met (465.6 µmol/L). Due to the suspected diagnosis, treatment with betaine, vitamin B_12_, vitamin B_6_, folic acid and a Met-restricted diet with 20 mg Met/kg/day was initiated. Plasma vitamin B_12_ levels were normal in the mother. On day 26, Met concentrations increased to a maximum level of 2271 µmol/L with only moderately elevated levels of tHcy of 59.7 μmol/L, suggestive of a MAT I/III deficiency. Betaine treatment was stopped immediately. Sequencing of *MAT1A* revealed a homozygous pathogenic variant, c.1132G>A; p.Gly378Ser. Met restriction (66% comida-HCys A FORMULA, Dr. Schär, Rosbach, Germany; 33% HiPP PRE BIO formula food, Pfaffenhofen, Germany) was continued with a target Met concentration < 500–600 µmol/L. Met concentrations under this dietary regimen are displayed in Figure 1.

At 5 months, a brain MRI revealed mildly delayed myelination, dilated inner and outer cerebrospinal fluid spaces and mild periventricular diffusion interference (Figure 2). Clinically, the patient showed mild, predominantly motor, developmental delay (no hand support possible in prone position; no age-appropriate head posture; no turning of the child in prone and/or supine position) and mild muscular hypotonia. From 5 to 8 months of age, weight stagnation—likely due to decreased caloric intake—was observed. Dietary adaptation with an age-adequate calorie intake resulted in catch-up growth and normal weight at 14 months of age. Psychomotor development and muscle tone normalized.

Psychomotor development at the age of 21 months was age-appropriate, and the patient showed normal growth under the Met-restricted diet (25 mg Met/kg/day). Met concentrations remained below 600 μmol/L. The concentration of AdoMet was normal (95.9 nmol/L, reference range 71–118 nmol/L). Despite the lack of neurological symptoms, a brain MRI at 21 months showed progressive diffusion impairment, but no structural abnormalities.

### 3.2. Analysis of the MAT1A Protein Structure

MAT1A is a protein of 395 amino acids that catalyzes the transfer of an adenosyl group from ATP to Met to produce AdoMet [13]. A discrete binding site for AdoMet has been described in the structure of human MAT1A (Figure 3A, PDB 2OBV) [12]. Hepatic MAT1A can adopt dimeric as well as tetrameric states. The X-ray crystal structure of human MAT1A shows that it is arranged as a dimer of dimers, with a bound AdoMet molecule at each active site located at the dimer interface. Thus, the AdoMet molecule makes contacts with both subunits of the dimer and it is structurally stabilized within the solvent-exposed microenvironment by the presence of a ‘gating loop’ [12].

### 3.3. Predicted Effect of the Homozygous c.1132G>A; p.Gly378Ser Mutation on the Enzymatic Activity of MAT I/III

The c.1132G>A variant of the patient is located at the C-terminus of MAT1A and is proposed to impair protein stability by β-turn disruption (Figure 3B) [12]. Studies performed with bacterial MAT showed that this enzyme is a substrate for the chaperone system GroEL GroES in bacteria [14], i.e., folding and maturation require chaperone-mediated stabilization. A previous study with human recombinant MAT1A Gly378Ser expressed in *Escherichia coli* (*E. coli*) and COS-1 cells showed a residual enzymatic activity of 0.17% compared to full-length intact MAT1A [15].

## 4. Discussion

MAT I/III deficiency is a rare inborn error of Met metabolism with less than 70 cases reported to date [4]. Little is known about the long-term outcome of this disorder and the necessity of treatment. In 2015, Chien et al. published an overview on the worldwide available data on patients with either homozygous or compound heterozygous mutations in *MAT1A* [4]. Of 64 patients included in this study, 32 showed CNS involvement (based mainly on MRI findings). The authors could show that CNS symptoms correlated with Met concentrations in plasma. Individuals without neurologic symptoms typically had mean Met levels below 800 μmol/L, while patients with CNS involvement had mean Met concentrations above 800 μmol/L. However, exceptions have been reported in both groups [4]. Thus, treatment targets for dietary management are hard to establish, and reducing methionine concentration below <800 µmol/L will probably not prevent symptoms in all patients. In our patient, who initially had maximum methionine concentrations above 2000 µmol/L under betaine treatment, methionine levels were successfully kept below 600 µmol/L under dietary treatment during the further course. Extremely elevated Met concentrations may lead to brain edema. Several cases have been reported among those patients with CBS deficiency under betaine treatment [16,17] and normal children with extreme hypermethioninemia (up to 6830 μmol/L) after ingestion of high Met-containing formula [2,18]. Only one patient with MAT I/III deficiency and brain edema under treatment with betaine has been described so far (Met concentrations between 960 and 1560 μmol/L) [16]. Our patient did not develop brain edema despite extreme hyperpemethioninemia of >2000 µmol/L during the neonatal period. In light of the risk of cerebral edema, the benefit of initiating treatment with betaine before diagnostic confirmation is unclear.

The pathophysiology of CNS symptoms is still not fully understood, and several factors have been discussed: (1) The toxicity of very high methionine levels themselves; (2) Deficiency of AdoMet due to a greater loss of MAT I/III activity; (3) Increases in tHcy (although homocysteine levels in MAT I/III patients are usually not higher than in treated CBS deficiency patients, who typically have normal IQs) [11]; and (4) A combination of more than one of these factors [4]. Our patient showed mild motor developmental delay and muscular hypotonia within the first year of life, but her psychomotor development and muscle tone normalized with age and has been consistently age-appropriate until today. Despite that, CNS abnormalities, in particular diffusion impairment, were slightly progressive. Similar cases with amelioration or normalization of neurologic symptoms with age have been reported previously [4].

While there is strong evidence that heterozygotes that are usually identified by NBS with mild to moderate hypermethioninemia will be clinically unaffected and do not require treatment [4,8,11], treatment recommendations for severely affected patients are less straightforward. Based on the available evidence, Chien et al. have proposed dietary treatment in patients with mean Met levels >500–600 µmol/L [4]. Met concentrations under treatment should not be lowered too much to avoid secondary AdoMet depletion [10,19]. Supplementation of AdoMet should be considered in patients with MRI changes, as the available evidence suggests that AdoMet supplementation can result in normalization of even established MRI abnormalities [2,6]. No clear genotype–phenotype correlation has been established in MAT I/III deficiency. Patients with homozygous truncating mutations with presumably no residual MAT I/III activity are found among those without CNS involvement, while most of the patients with neurologic symptoms only carry missense mutations with some residual MAT activity [4]. Studies by Chamberlin on brain demyelination due to MAT I/III deficiency revealed that patients with missense mutations that cause partial inactivation of the enzymatic activity are usually free from severe clinical complications [20]. The authors observed that MAT I/III mutations that lead to truncated variants of the protein, devoid of enzymatic activity, are associated with a greater risk for the development of brain demyelination [20]. Interestingly, brain demyelination in patients carrying a truncated variant of MAT I/III may present very late in infancy, with two cases presenting CNS involvement only at the age of 11 years old [20]. The authors suggested that, because myelination is a long-term process, compensatory biosynthesis of the myelin precursors phosphatidylcholine and spinghomyelin may be supported by AdoMet pools derived from MAT II (primarily extra hepatic), and this supply may be enough to prevent early-onset CNS symptoms in the patient. The possibility of choline and AdoMet supplementation appears relevant to prevent demyelination in patients with truncated variants of MAT I/III. Our patient carries a homozygous missense mutation, a type of mutation that often escapes mRNA nonsense-mediated decay (NMD) [21], leading to protein expression with a residual activity of 0.17%. For all genetic diseases, patients with premature termination codons (PTCs), namely, nonsense and frameshift mutations, often present with milder phenotypes as compared to missense mutations [21]. This could be explained by the fact that most PTCs are captured by the NMD system. This lowers mRNA expression, thereby preventing or markedly reducing the expression of the abnormal protein, which could engage in intracellular protein aggregation and cytotoxicity. The c.1132G>A (p.Gly378Ser) variant that was found in homozygosity in our patient has previously been described in three compound heterozygous individuals, among those, one pair of siblings [22]. None of these patients showed CNS abnormalities, but no MRI was performed [5]. The c.1132G>A variant is located at the C-terminus of MAT1A and is proposed to impair protein stability by β-turn disruption [12], resulting in a reduced MAT I/III activity of 0.17% when expressed in *E.coli* and COS1-cells. In our patient, the likely normal MAT1A expression, the extremely low residual enzymatic activity and the fact that choline uptake by the brain reduces with age [23] may compromise the long-term availability of precursors for myelin biosynthesis and remodelling, and thus follow up of the CNS status is recommended.

## 5. Conclusions

The case of our patient demonstrates that the neurological symptoms of MAT I/III deficiency can be transient and even patients with neurologic manifestations can show normal development under a low-methionine diet. This is especially interesting as the patient harbors a homozygous mutation with almost no residual activity.

Further studies are needed to better characterize the long-term outcome of this rare disorder and the necessity of treatment.

## Figures and Tables

**Figure 1 genes-13-01163-f001:**
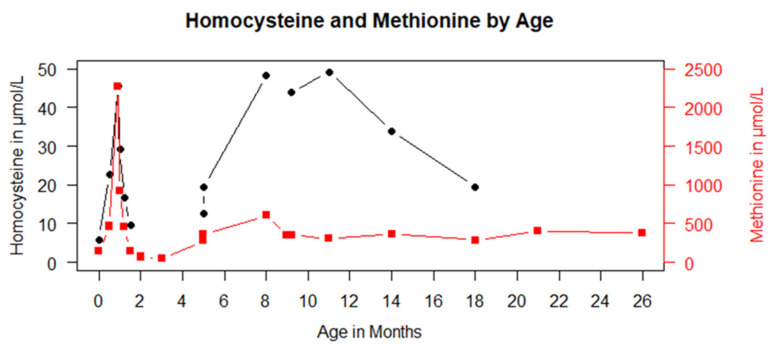
Methionine (red) and total homocysteine (black) concentrations are plotted against age in months.

**Figure 2 genes-13-01163-f002:**
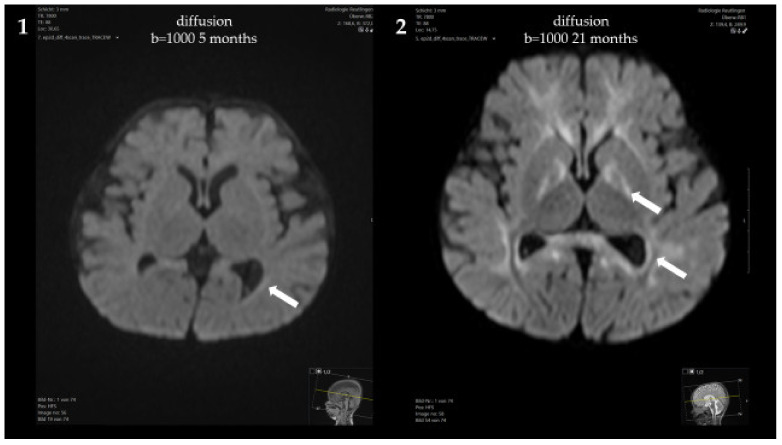
Brain MRI of the patient: (**1**,**3**) At 5 months, showing mildly delayed myelination, dilated inner and outer cerebrospinal fluid spaces and mild periventricular diffusion interference; (**2**,**4**) At 21 months, showing progressive diffusion impairment.

**Figure 3 genes-13-01163-f003:**
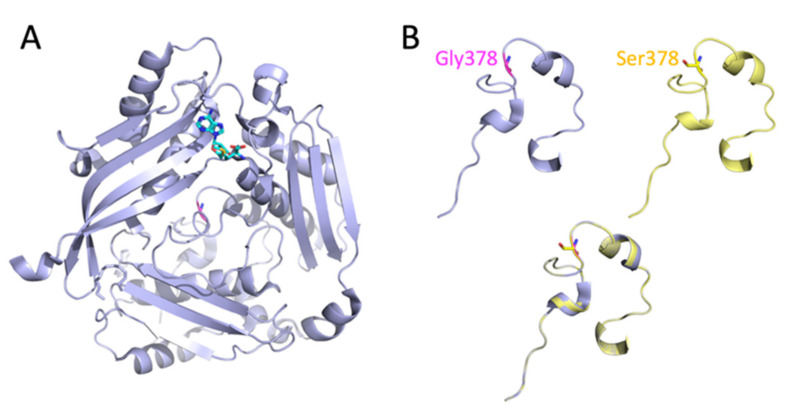
Structure of monomeric human MAT1A. (**A**) Monomeric MAT1A (PDB 2OBV) is shown as light blue ribbons, with AdoMet represented as sticks shown in cyan. Target residue Gly378 is located in the C-terminal region of MAT1A within loops and β-turns, and is herein shown as magenta sticks. (**B**) Close up of the location of Gly378 within the C-terminus of MATA1 in wild-type MAT1A (light blue), results from in silico mutagenesis to Ser (yellow), and alignment of native (Gly378) and mutant (Ser378) MAT1A proteins (overlay, light blue and yellow).

## Data Availability

Not applicable.

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
