# Peer review of "Methionine Adenosyltransferase I/III Deficiency Detected by Newborn Screening"

_genes, 2022, doi:10.3390/genes13071163_

Round 1

Reviewer 1 Report

The authors provide a clear description of the presented case, with all necessary information.

The introduction to the disorder described is also complete and conclusive.

The discussion is sound and conclusive.

The authors present a case report of patient with MAT I/III deficiency detected by newborn screening. They have followed the patient over more than 2 years, and they present all relevant data like, methionine and homocysteine values, data on protein restriction, and brain MRI at the age of 5 and 21 months, showing the course and progression of blood levels of metabolites, and the delay of myelination. The authors have thoroughly searched the literature, and key publications are referred to and cited in an appropriate way. In addition the authors have also investigated and described the effect of the homozygous mutation in the MAT I/III gene. In the discussion the authors have put together the data and disease course of their patient and compared it to already described cases and recommendations. Finally the authors have also discussed the inconsistency of existing data and recommendation, regarding prognosis of disease course and severity, recommendation of when and how to treat patients with MAT I/III deficiency. This paper will add a lot to the knowledge of MAT I/III deficiency, and although it does not resolve or answer all existing questions, it will help to focus on further necessary research.

Reviewer 2 Report

This is a well-written manuscript describing a rare case of methionine adenosyltransferase I/II.

Line 58: Does Genes prefer decimal instead of comma for thousands?

Line 114 – spelling of calory/calorie